

# Intake of supplementary food during pregnancy and lactation and its association with child nutrition in Timor Leste

Sanni Yaya[1,*], Ruoxi Wang[2,*], Shangfeng Tang[2] and Bishwajit Ghose[1,3]

[1] School of International Development and Global Studies, University of Ottawa, Ottawa, ON, Canada
[2] School of Medicine and Health Management, Tongji Medical College, Wuhan, Hubei, China
[3] Institute of Nutrition and Food Science, University of Dhaka, Dhaka, Bangladesh
* These authors contributed equally to this work.

Corresponding author
Bishwajit Ghose,
brammaputram@gmail.com

## ABSTRACT

**Abstract:** There is little evidence on maternal consumption of supplementary food on nutritional status of children. The objectives of this study were to measure the prevalence and determinants of supplementary food intake during pregnancy and lactation, and their association with nutritional status of under-five children in Timor Leste.
**Methods:** Cross-sectional data from Timor Leste Demographic and Health Survey on 5,993 mother (15–49 years) child dyads (<5 years) were included in the analysis. Self-reported intake of supplementary food intake was the explanatory variable. Child's nutritional status was assessed by stunting, wasting, and underweight and categorized according to WHO recommendations.
**Results:** The prevalence of taking supplementary food during pregnancy and lactation was, respectively, 29.1% (95% CI [27.2–31.0]) 31.0% (95% CI [29.1–33.0]), and that of taking iron supplement during pregnancy was close to three-fifths (63.1%, 95% CI [60.9–65.3]). The odds of taking supplementary food during pregnancy and lactation were lower among those in the younger age groups and higher among urban residents. Compared with mothers who had supplementary food during pregnancy and lactation, those did not have had respectively 1.36 (OR = 1.360, 95% CI [1.191–2.072]) and 1.15 times (OR = 1.152, 95% CI [1.019–1.754]) higher odds of having stunted, and 1.30 (OR = 1.307, 95% CI [1.108–1.853]) and 1.43 (OR = 1.426, 95% CI [1.140–1.863]) times higher odds of having underweight children. Those who had none of the supplements had respectively 1.67 (OR = 1.674, 95% CI [1.118–2.087]) and 1.63 (OR = 1.631, 95% CI [1.130–2.144]) times higher odds of having stunted and underweight children.
**Conclusion:** A great majority of the mothers in Timor Leste are not taking supplementary food during pregnancy and lactation. We found a positive relationship between supplementary food intake during pregnancy and lactation with stunting and wasting among under-five children.

## INTRODUCTION

Regular consumption of a varied and balanced diet from preconception through pregnancy and post-pregnancy period is of paramount importance to ensure healthy

birth outcomes as well as optimum nutritional status of the mother and her newborn (*Abu-Saad & Fraser, 2010*; *Marangoni et al., 2016*). Maternal undernutrition/inadequate dietary intake is strongly correlated with poor growth of the fetus in utero and post-nataly with demonstrated enhanced susceptibility for developing chronic diseases, for example, high blood pressure, type-2 diabetes, and neurological disorders in the later stages of life (*Belfort et al., 2008*; *Wen et al., 2011*; *Brenseke et al., 2013*; *Keenan et al., 2013*; *Szostak-Wegierek, 2014*; *Blumfield et al., 2015*). Various metabolic adaptations that take place to meet the increased nutritional demand involve hormonal mechanisms that play crucial roles in regulating appetite and eating behavior (*Hirschberg, 2012*). Although pregnancy embodies many of the factors that are known to trigger hyperphagia/overeating (*Van Der Wijden et al., 2014*; *Orloff et al., 2016*), exceptions are also widespread as a lot of women experience frequent bouts of food aversions and anorexia nervosa (*Bayley et al., 2002*; *Dinas et al., 2008*; *Bulik et al., 2009*) and share higher risks of miscarriage, intrauterine growth restriction, still birth, low birthweight (*Koubaa et al., 2005*; *Bulik et al., 2009*). The situation is particularly complex for women who are already malnourished and enter pregnancy with certain nutritional deficiencies.

In low-income settings, maternal malnutrition during pregnancy and lactation is a common phenomenon and is most often associated with inadequate intakes of specific micronutrients, resulting in adverse birth and health consequences for the child, for example, preterm birth due to zinc deficiency, and low birth weight due to iron/folic acid deficiency, cretinism due to iodine deficiency (*Abu-Saad & Fraser, 2010*; *Gernand et al., 2016*). To account for the rising nutritional demand during pregnancy and lactation, women attending antenatal care are encouraged to undergo nutritional assessment, given dietary counselling, and routinely prescribed to take supplementary food enriched in micronutrients to meet the recommended dietary allowances (RDAs) (*Tsui, Dennehy & Tsourounis, 2001*; *Khanal, Zhao & Sauer, 2014*; *Abdullahi et al., 2014*). Setting RDAs for various micronutrients for pregnant mothers is challenging as it is dependent on a host of physiological factors. However, RDAs determined by local experts in nutrition such as registered dietitians are specific in guiding intakes for women in accordance with their nutritional status, dietary behavior, cultural preferences to ensure optimum pre- and post-natal nutrition for mother and her child. Despite the existing research base on dietary guidelines and maternal nutrition promotion programs, achieving optimum nutritional status continues to be a challenging task especially in low-income settings due to lack of nutrition education, access to appropriate food, and diverse sociodemographic structure of the population (*Lowdon, 2008*).

Apart from maintaining a balanced diet, encouraging the intake of supplementary food has also been a popular strategy for addressing the problem of micronutrient deficiency among pregnant women. However, assessing the beneficial effects of maternal supplementation programs on fetal and childhood nutrition is complex owing to the interplay of a range of behavioral, environmental, psychosocial factors that vary widely across cultures (*Abu-Saad & Fraser, 2010*). Also, the impacts of such programs may not be universal as studies have produced mixed outcomes across and within countries, implying the influence of population contextual factors in the responsiveness of different

supplementation programs. From this perspective, assessing the relationship between supplementary food intake at various stages of pregnancy and lactation on a child's nutritional status carry special importance as it provides the basis for making effective nutrition policies targeting the contextual situation and health needs of a population. To this end, we conducted this study based on a national survey in Timor Leste, a young and fragile state located in Southeast Asian region which has so far received very little attention on health research globally (*Deen et al., 2013*).

As is the case in many other developing economies of Asia, poverty and undernutrition remain the main contributors to maternal and child mortality in Timor Leste, presenting an unparalleled challenge for human development in the country (*Haddad, Philpott-Jones & Schonfeld, 2015*). Since its independence in 2002, the government has made several initiatives to fight malnutrition among pregnant women such as free provision of iron supplementation among pregnant women. According to the nutrition strategy, all pregnant women are supplied with iron/folate acid tablets free of charge. Information on supplementary food intake were not collected in the latest Demographic and Health Survey (DHS), therefore we used data from round-2 (2009–2010) to assess the relationship between supplementary food intake during pregnancy and lactation and nutritional status among children, as there is currently no research evidence on the effectiveness of such practice in Timor Leste. The findings of this study are expected to be of great value to the ongoing maternal and child nutrition programs in Timor Leste, as well as other countries in the region sharing similar sociodemographic characteristics.

## METHOD

### Setting

Democratic Republic of Timor Leste, also known as East Timor or Timor Leste, is the newest country located in the Asia Pacific region, and also one of the smallest in Asia in terms of total population. The country declared independence from Indonesia in 2002, after a prolonged period of conflict and destruction of economic and health infrastructures. As one of the least-developed countries, Timor Leste relies heavily on foreign aid for supporting its development agenda. The population is predominantly rural and engaged in subsistence agriculture. Median age of the population is 17.5 years with young people (0–14 years) constituting about 40% of the population (*Timor-Leste Age structure—Demographics, 2018*). The demographic is characterized by high fertility (almost six children per women), and high maternal mortality rates (557 deaths per 100,000 deliveries) and under-five (64 deaths per 1,000 births) mortality rates (*Richards, 2015*; *Price et al., 2016*). Survey description: data for this study were extracted from the second round of Timor Leste Demographic and Health Survey (TLDHS) conducted in 2009–2010. The main objectives of the DHS surveys are to generate quality data on key sociodemographic indicators (e.g., maternal and child malnutrition, maternal healthcare use, HIV knowledge), monitoring national and international health related goals, and to assist in evidence-based policy making in the host countries. The survey was implemented by the National Statistics Directorate of the Directorate

General for Analysis and Research of the Ministry of Finance with technical assistance from ICF Macro, and financial support by several international donors such from USAID and WHO.

The sampling process involved a two-stage selection strategy. Firstly, the selection of clusters from the 13 districts in both urban and rural areas, and secondly to select households from each cluster for interview. In total 455 clusters were selected (116 urban areas and 339 rural areas) from which 11,800 women were eligible for interview. Field work lasted from August 10, 2009 to February 7, 2010. The survey methods were published with more details elsewhere (*DHS Program, 2016*)

## Variables used in the study

### Outcome variable

Outcome variables were three measures of child malnutrition: height-for-age (stunting), weight-for-age (underweight), and weight-for-height (wasting). These are commonly used and reliable indicators of nutritional status that provide information regarding growth and body composition among children under 5 years of age. These are measured in standard deviation units ($z$-scores) from the median of the reference population, as recommended by WHO (*Stevens et al., 2012*).

Stunting is an indication of linear growth retardation associated with poor feeding practices, chronic inadequate intake of protein and energy and frequent infection. It was categorized as: stunted (height for age: HFA < −2 SD), and not stunted (HFA ≥ −2 SD).

Wasting is an indication of acute undernutrition resulting from of insufficient food intake, failure to gain weight or weight loss, recurrent infectious diseases (e.g., diarrhea), or a combination of these factors. It was categorized as: wasted (weight for height: WFH < −2 SD), and not wasted (WFH ≥ −2 SD).

Underweight is a composite indicator of stunting and wasting that accounts for both acute and chronic malnutrition. It was categorized as: underweight (weight for age: WFA < −2 SD), and not underweight (WFA ≥ −2 SD).

### Independent variables

The main explanatory variables were self-reported intake of supplementary food during pregnancy and lactation. These were assessed by asking the participants: (1) whether or not they took any supplementary food during their most recent pregnancy (Yes/No), (2) whether or not they took any supplementary food during lactation for their most recent child.

### Control variables

To measure the independent association between the outcome and explanatory variables, the analysis was adjusted for a generous number of maternal and child level variables which are likely to influence nutritional status of children. The variables are listed and described in Table 1.

**Table 1 Maternal demographic and socioeconomic characteristics (_n_ = 5,993).**

|  | Variable description | (%) | 95% CI (lower–upper) |
|---|---|---|---|
| **Maternal-level variables** |  |  |  |
| Residency type | Urbanicity of the place of residence |  |  |
| Urban |  | 24.6 | 23.1–26.3 |
| Rural |  | 75.4 | 73.7–76.9 |
| Religion | Religious affiliation of the respondent |  |  |
| Roman catholic |  | 98.0 | 97.3–98.5 |
| Other |  | 2.0 | 1.5–2.7 |
| Education | Educational level attained based on number of formal schooling years |  |  |
| Nil |  | 32.9 | 31.0–34.8 |
| Primary |  | 27.5 | 26.0–29.1 |
| Secondary/higher |  | 39.6 | 37.2 42 |
| Parity (mean) | Number of children ever born | 4.40 | 4.33 4.46 |
| 1–2 |  | 28.8 | 27.3–30.3 |
| 3–4 |  | 28.4 | 27.1–29.7 |
| >4 |  | 42.8 | 41.2–44.5 |
| BMI | Individual's weight in kilograms (kg) divided by height in meters squared ($m^2$) |  |  |
| Normal weight |  | 23.7 | 22.4–25.1 |
| Overweight |  | 68.3 | 66.7–69.9 |
| Obese |  | 8.0 | 7.1–9.1 |
| Household wealth status | Wealth status assessed based on wealth quintile: lowest/lower quintile = poor, and middle/higher/highest = non-poor |  |  |
| Poor |  | 39.9 | 37.5–42.2 |
| Non-poor |  | 60.1 | 57.8–62.5 |
| During pregnancy had supplementary food | Self-reported status of supplementary food intake during pregnancy |  |  |
| No |  | 70.9 | 69.0–72.8 |
| Yes |  | 29.1 | 27.2–31.0 |
| During lactation had supplementary food | Self-reported status of supplementary food intake during lactation |  |  |
| No |  | 68.9 | 67.0–70.8 |
| Yes |  | 31.0 | 29.1–33.0 |
| During pregnancy had iron supplements | Self-reported status of iron supplementation intake during pregnancy |  |  |
| No |  | 36.9 | 34.7–39.1 |
| Yes |  | 63.1 | 60.9–65.3 |
| **Child-level variables** |  |  |  |
| Sex | Sex of the child |  |  |
| Male |  | 51.9 | 50.4–53.4 |
| Female |  | 48.1 | 46.6–49.6 |
| Birth type | Whether or not birth was singleton |  |  |
| Singleton |  | 99.0 | 98.7–99.2 |
| Twin |  | 1.0 | 0.8–1.3 |

| | Variable description | (%) | 95% CI (lower–upper) |
|---|---|---|---|
| **Skilled birth** | Birth took place at home or a health facility | | |
| No | | 74.8 | 72.8–76.8 |
| Yes | | 25.2 | 23.2–27.2 |
| **Had early initiation of BF** | Breastfeeding started within 1 h of birth | | |
| No | | 9.90 | 9.0–10.9 |
| Yes | | 90.1 | 89.1–91.0 |
| **Taking iron supplements** | Whether or not currently taking iron pills/sprinkles/syrup | | |
| No | | 79.5 | 78.2–80.8 |
| Yes | | 20.3 | 19.0–21.6 |
| **Stunted** | Height/age below < 2 SD | | |
| Yes | | 36.7 | 35.1–38.3 |
| No | | 63.3 | 61.7–64.9 |
| **Underweight** | Weight/age below < 2 SD | | |
| Yes | | 36.0 | 34.5–37.7 |
| No | | 64.0 | 62.3–65.5 |
| **Wasted** | Weight/height below < 2 SD | | |
| Yes | | 12.5 | 11.5–13.5 |
| No | | 87.5 | 86.5–88.5 |

## Data analysis

Data analysis was performed using SPSS V. 24. The dataset was first checked to make sure it was free from outliers and collinearity issues. To meet the requirements of the study, mothers who were currently breastfeeding and responded the questions on supplementary food intake were included in the analysis. The prevalence of stunting, wasting, and underweight, along with maternal and child level variables were presented as percentages with 95% confidence intervals. Following that, the prevalence of each of the outcome variables according to the intake of supplementary food during pregnancy and lactation were presented as bar charts. The final part of the analysis constituted a multivariable regression analysis to calculate the odds of association between stunting, wasting, undernutrition with the intake of supplementary food in three steps. The first step was univariate analysis (Model-1) without including any control variable in the model. In the second model the association was adjusted for maternal level variables (Model-2), while the last model adjusted for both maternal and child level variables (Model-3) to observe the differential contribution of these factors (*Ghose et al., 2016*). Maternal and child level variables were entered in different steps to find changes in Odds ratio for respective factors. Level of significance was set at $p < 0.05$ for all analyses.

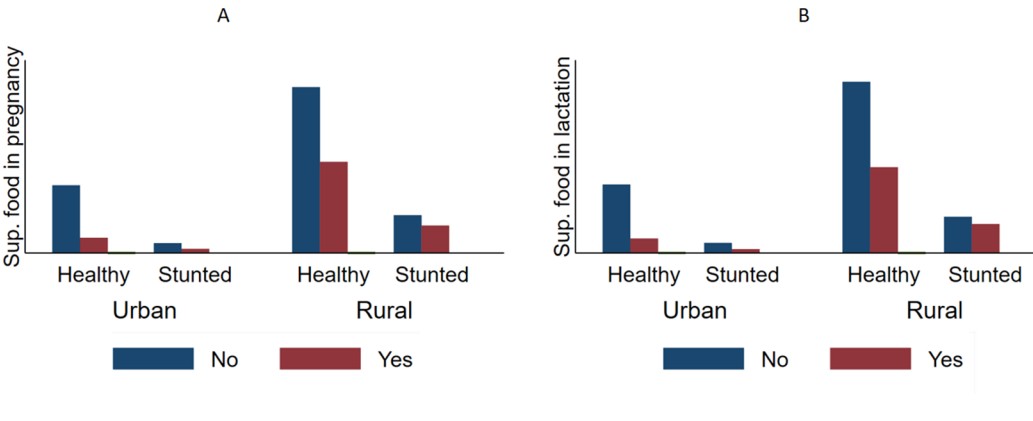

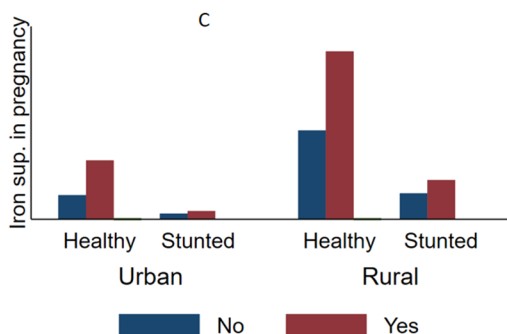

**Figure 1 Comparative prevalence of stunting by supplementation intake during pregnancy and lactation stratified by type of residency in Timor Leste (2009–2010).** The prevalence of stunting (HFA < 2 SD) according the supplementary food intake status among urban and rural mothers. The figure indicates that mothers who took supplementary food during pregnancy (A) and lactation (B) had lower likelihoods of having stunted children compared with those who did not. This was true in both urban and rural areas. Taking iron supplementation (C) appeared to have a positive relationship with stunting; however, the difference was not statistically significant.

## Ethical approval

All DHS survey protocols are approved by ICF international and a review board in the country of survey. Data were made available in the public domain in anonymized form. Therefore, no additional approval is necessary.

## RESULTS

### Descriptive statistics

The study participants were 5,993 mothers and their last-born child. The mean age of mothers and children was respectively 31.53 (31.34–31.72) and 1.32 (1.29–1.35) years. The sociodemographic characteristics of the mothers and children are presented in Table 1.

### *Prevalence of women who received supplementary food during pregnancy and lactation*

Less than one-third of the mothers reported taking supplementary food during pregnancy (29.1%, 95% CI [27.2–31.0]) and lactation (31.0%, 95% CI [29.1–33.0]),

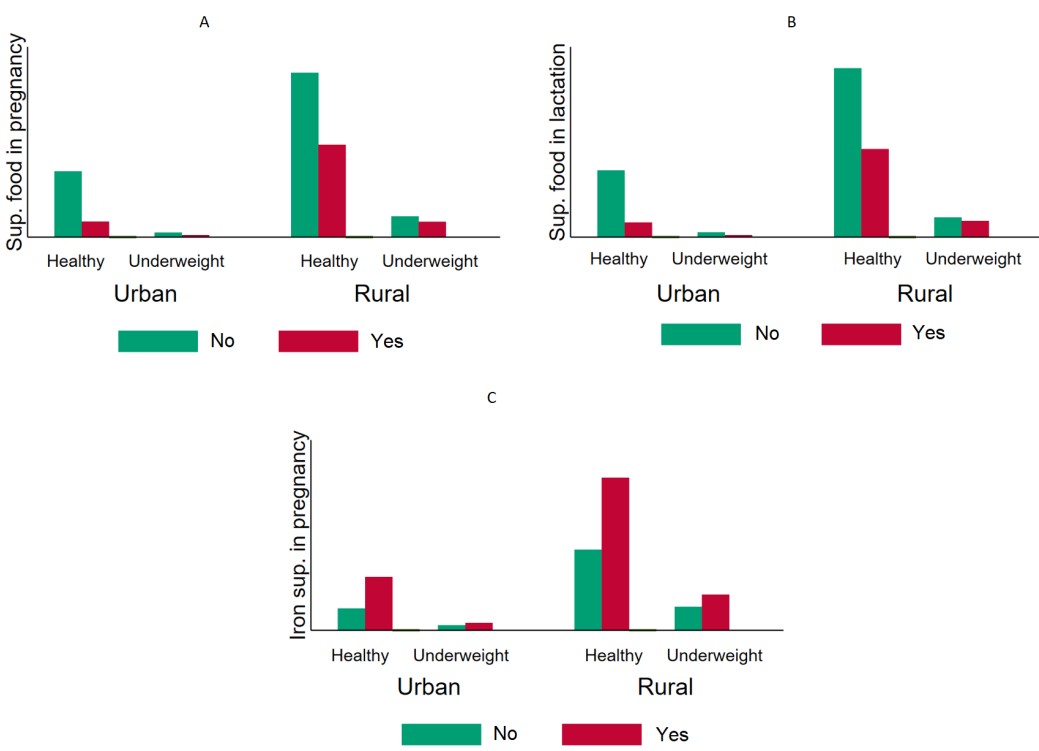

**Figure 2 Comparative prevalence of underweight by supplementation intake during pregnancy and lactation stratified by type of residency in Timor Leste (2009–2010).** The overall prevalence of having underweight (WFA < 2 SD) children was higher among mothers who did not take supplementary food during pregnancy (A) and lactation (B). The same pattern was observed when the sample was stratified into urban and rural areas. In the chi-square tests, iron intake (C) did not show any significant decrease in the probability of being underweight.

whereas the prevalence of taking iron supplement was close to three-fifths (63.1%, 95% CI [60.9–65.3]).

### Prevalence of stunting, wasting, and underweight

Table 1 also shows that more than one-third of the children were stunted (36.7%, 95% CI [35.1–38.3]), underweight (36.0%, 95% CI [34.5–37.7]), and 12.5% (95% CI [11.5–13.5]) were wasted (Figs. 1–3).

### Sociodemographic correlates of taking supplementary food among pregnant and lactating mothers

Table 2 shows the results of the multivariable regression on the association between supplementary food intake with the sociodemographic characteristics of the participants. The odds of taking supplementary food during pregnancy and lactation were lower among those in the younger age groups (20–34 years), higher among those in the urban areas, had 1–2 children. Surprisingly, women who had no education (OR = 1.416, 95% CI [1.104–2.011]) and lived in the poor households (OR = 1.427, 95% CI [1.024–1.911]) had higher odds of taking iron supplements during pregnancy. The odds of taking two or all three supplements were also higher among urban mothers compared with those in the rural areas.

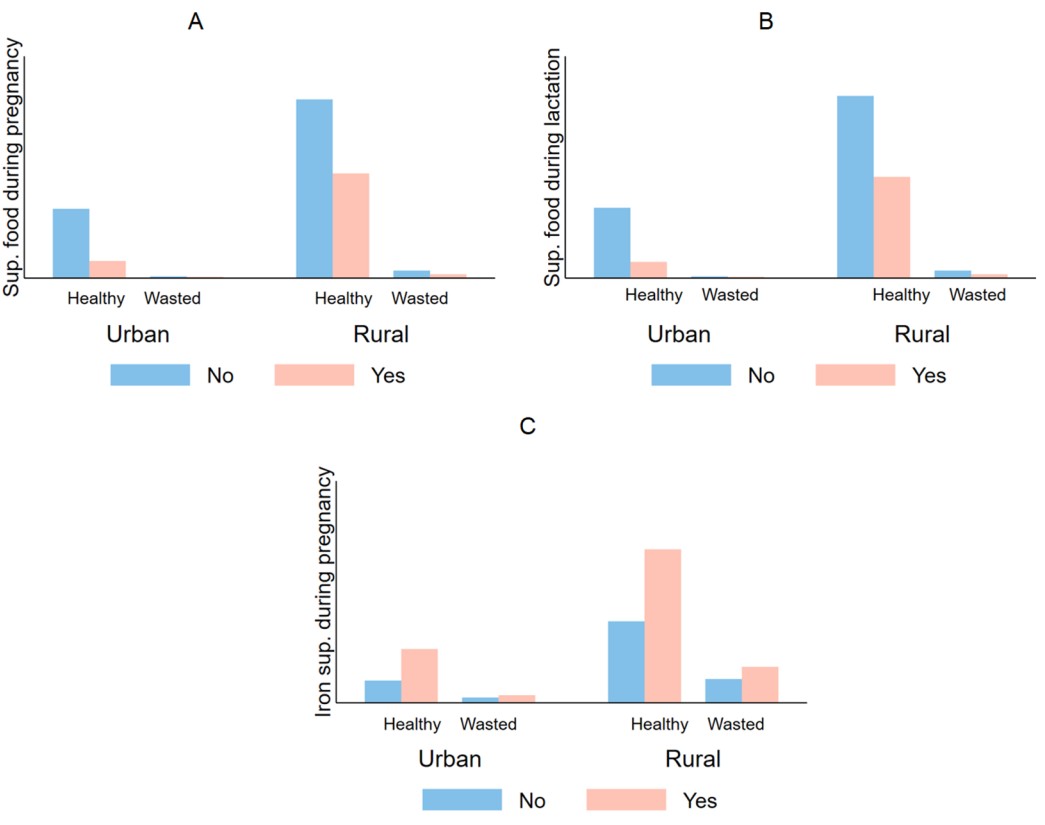

**Figure 3 Comparative prevalence of infant wasting by iron supplementation during pregnancy and lactation stratified by type of residency in Timor Leste (2009–2010).** In both urban and rural areas mothers who did not take supplementary food during pregnancy (A) and lactation (B), iron supplementation during pregnancy (C), had a higher prevalence of having babies who are wasted (WFH < 2 SD).

### Multivariable association between supplementation intake during pregnancy and lactation and child nutritional outcomes

Table 3 summarizes the results (odds ratios) of the association between supplementation intake during pregnancy and lactation with child stunting, underweight, and wasting. In general, compared with mother who reported taking supplementary food during pregnancy and lactation, those who did not take had higher odds of children with stunting, underweight, and wasting. The same was true for those who took supplementary food both during pregnancy and lactation, and took all three types supplementation compared with those who took none or only one (with a few exceptions). For instance, compared with mothers who had supplementary food during pregnancy and lactation, those did not have had respectively 1.61 (OR = 1.610, 95% CI [1.128–2.310]) and 1.57 (OR = 1.570, 95% CI [1.108–2.127]) times higher odds of having stunted and underweight children (Model-1). Those who had none of the supplements at any point had respectively 1.67 (OR = 1.674, 95% CI [1.118–2.087]) and 1.63 (OR = 1.631, 95% CI [1.130–2.144]) times higher odds of having stunted and underweight children. The strength of these associations decreased to some extent upon adjustment for the

Table 2 Odds ratio of taking supplementary food during pregnancy and lactation in Timor Leste.

| | During pregnancy had supplementary food | During lactation had supplementary food | During pregnancy had iron supplements | Two supplementations* | Three supplementations** |
|---|---|---|---|---|---|
| Age groups (45–49) | | | | | |
| 15–19 | 0.683[ns] | 0.634[ns] | 0.996[ns] | 0.704[ns] | 0.796[ns] |
| 20–24 | 0.585 | 0.510 | 0.904[ns] | 0.514 | 0.744[ns] |
| 25–29 | 0.706 | 0.617 | 0.801[ns] | 0.639 | 0.778[ns] |
| 30–34 | 0.704 | 0.619 | 0.767[ns] | 0.640 | 0.867[ns] |
| 35–39 | 0.914[ns] | 0.764[ns] | 0.785[ns] | 0.857[ns] | 10.057[ns] |
| 40–44 | 0.891[ns] | 0.657 | 0.906[ns] | 0.801[ns] | 0.950[ns] |
| Residency type (rural) | | | | | |
| Urban | 4.168 | 4.006 | 0.548[ns] | 4.531 | 3.599 |
| Religion (other) | | | | | |
| Roman catholic | 1.590 | 0.861[ns] | 1.052[ns] | 1.016[ns] | 1.284[ns] |
| Education (secondary/higher) | | | | | |
| Nil | 0.892[ns] | 0.877[ns] | 1.416 | 0.887[ns] | 0.958[ns] |
| Primary | 0.909[ns] | 0.898[ns] | 0.968[ns] | 0.885[ns] | 0.846[ns] |
| Parity (>4) | | | | | |
| 1–2 | 1.153 | 1.193 | 0.856[ns] | 1.352 | 1.267 |
| 3–4 | 1.000[ns] | 0.991[ns] | 0.840[ns] | 1.025[ns] | 0.950 |
| BMI | | | | | |
| Normal weight | 0.751[ns] | 0.679 | 0.722[ns] | 0.731 | 0.574[ns] |
| Overweight | 0.845[ns] | 0.861[ns] | 0.857[ns] | 0.831[ns] | 0.722[ns] |
| Household wealth status (non-poor) | | | | | |
| Poor | 1.089[ns] | 1.144[ns] | 1.427 | 1.176 | 1.032[ns] |

Notes:
ns = Not significant ($p > 0.05$).
* Two supplementations = During pregnancy had supplementary food + During lactation had supplementary food.
** Three supplementations = During pregnancy had supplementary food + During lactation had supplementary food + During pregnancy had iron supplements.
(Reference) categories in parenthesis.

maternal (Model-2) and child level variables (Model-3), however, remained statistically significant (except for wasting).

## DISCUSSION

### Main findings

Findings of the present study based on the data from DHS revealed a considerably high prevalence of undernutrition, especially of stunting and wasting among under-five children in Timor Leste. Characteristic of post-conflict communities, child malnutrition is rampant across the country and represents a serious public health concern. What is perhaps of greater concern is the notably low prevalence of taking supplementary food among women during pregnancy and lactation. Dietary supplementation during pregnancy may not be considered as an absolute necessity for a high income setting with stable provision of food and livelihood amenities. However, the current situation in Timor Leste characterized by a fragile agri-food sector and precarious state of food and

**Table 3 Odds ratios of child stunting, underweight, and wasting among mother who did not take supplementary food during pregnancy and lactation in Timor Leste (2009–2010).**

| | Model-1 | | | Model-2 | | | Model-3 | | |
|---|---|---|---|---|---|---|---|---|---|
| | Stunting | Underweight | Wasting | Stunting | Underweight | Wasting | Stunting | Underweight | Wasting |
| **During pregnancy had supplementary food (Yes)** | | | | | | | | | |
| No | 1.435 | 1.328 | $1.157^{ns}$ | 1.339 | 1.262 | $0.879^{ns}$ | 1.360 | 1.152 | $0.906^{ns}$ |
| **During lactation had supplementary food (Yes)** | | | | | | | | | |
| No | 1.709 | 1.672 | $1.130^{ns}$ | 1.603 | 1.589 | $1.042^{ns}$ | 1.307 | 1.426 | $1.061^{ns}$ |
| **During pregnancy had iron supplements (Yes)** | | | | | | | | | |
| No | $0.997^{ns}$ | $1.082^{ns}$ | $0.957^{ns}$ | $1.074^{ns}$ | $1.146^{n}$ | 1.198 | $1.025^{ns}$ | $1.120^{ns}$ | 1.226 |
| **Two supplementations* (Yes)** | | | | | | | | | |
| No | 1.61 | 1.570 | 1.138 | 1.487 | 1.491 | $1.253^{ns}$ | 1.221 | 1.334 | 1.130 |
| **Three supplementations**** | | | | | | | | | |
| No | 1.674 | 1.631 | $1.009^{ns}$ | 1.554 | 1.541 | 1.017 | 1.309 | 1.371 | $1.163^{ns}$ |

**Notes:**
ns = Not significant ($p > 0.05$).
Model-1, unadjusted; Model-2, adjusted for maternal level variables; Model-3, adjusted for maternal and child level variables.
* Two supplementations = During pregnancy had supplementary food + During lactation had supplementary food.
** Three supplementations = During pregnancy had supplementary food + During lactation had supplementary food + During pregnancy had iron supplements.
(Reference) categories in parenthesis.

nutrition insecurity (*Molyneux et al., 2012*) leave little doubt that micronutrient deficiency/ hidden hunger must also be widespread and contributing to the high rates of child malnutrition.

According to the findings, there remains certain sociodemographic patterns in the intake of supplementary food among pregnant and lactating mothers in the country. Mothers who were comparatively younger and residents of rural areas were less likely to take supplementary food. No significant difference in taking supplementary food was observed across socioeconomic groups (wealth status and education). The reasons why mothers of certain age groups and regions are more deprived of this facility than others remain subject to further exploration.

Overall, the findings of the study revealed a positive association between maternal consumption of supplementary food and nutritional status of children, reaffirming the role of maternal dietary practices on nutritional well-being of child. Apart from few exceptions, intake of supplementary food during pregnancy and lactation proved to have beneficial influence on nutritional status among Timorese children. Oddly enough, iron intake was not significantly associated with any of the outcomes. Iron deficiency (anemia) constitutes a leading nutritional deficiency among women of childbearing age especially in the developing countries, and is routinely included in the integrated antenatal care package (*Gautam et al., 2008*). Therefore, the findings should be interpreted with caution as we could not adjust the analysis for indicators of iron deficiency, as well as diseases that affect iron metabolism. There remains a scarcity of evidence on dietary supplementation during pregnancy and its impact on child nutritional outcomes as the studies have mostly been concerned with birth outcomes, which limits the scope of discussing the findings in contrast to past evidences. This is perhaps because birth

outcomes serve as strong predictors of nutritional status among under-five children. In Nepal, iron/folate supplementation among pregnant women were reported to reduce the risk of stunting among children below 24 months of age (*Ezeh et al., 2014*). The beneficial effects of dietary supplementation during pregnancy on birthweight were reported in several other countries with high rates of child malnutrition in Africa, for example, Ghana (*Adu-Afarwuah et al., 2015*), Kenya (*Maina-Gathigi et al., 2013*), Malawi (*Prado et al., 2016*), and Asia, for example, Bangladesh (*Mridha et al., 2016*; *Dewey et al., 2017*), India (*Aguayo et al., 2016*).

## General discussion and policy recommendation

Child malnutrition in Timor Leste stands at an alarming proportion, which is a potential indication of poor maternal health status and inadequate care and diet during pregnancy and lactation. Our findings show that majority of the women fail to take to advantage of taking supplementation during gestation. Important sociodemographic disparities were also observed in the prevalence of taking supplementary food. Being a country under the process of recovery from decades of political conflict and destruction of livelihood, the healthcare system in Timor Leste is faced with enormous challenges in terms of skilled human resource, inadequate research and development expenditures. These factors exert substantial cumulative effect on population health and nutrition, building a healthy workforce, and thereby can hamper socioeconomic development due to the cyclical nature of the relationship between poverty, health and national progress. In order to reverse this trend, the healthcare system need to make strategic planning and budget allocation for micronutrient supplementation and fortification programs that are proven cost-effective interventions for maternal and child malnutrition. Being a multi-ethnic population, uniform nutrition policy making may not address the contextual factors responsible for malnutrition and associated higher susceptibility to infectious diseases. We also observed a clear urban–rural disparity in the prevalence of supplementation food which indicates potential gaps in the coverage of the programs. Special attention is therefore necessary to address the regional gaps to make sure that all women can benefit from the programs regardless of their place of residence. Future studies should attempt to conduct qualitative studies and more nuanced analysis of sociocultural factors that influence nutritional practices such as use of supplementary/fortified food in the context of maternal and child malnutrition.

## Strengths and limitations

To our knowledge, this is the first study to report the association between dietary supplementation during pregnancy and lactation and nutritional status of children. Previous studies have mostly focused on the birth outcomes and considered supplementation during pregnancy. For this study, supplementation during both pregnancy and lactation were considered as poor nutrition among lactating mothers has been shown to be associated with adverse health outcomes among infants (*Keenan et al., 2013*). Besides that, the availability of large numbers of variables in the dataset allowed us to adjust the analysis for several maternal and child level variables.

The sample size was also high and the data was good quality. Among the limitations are the self-reported nature of several variables, for example, breastfeeding, and intake of dietary supplements. Also, the prevalence of stunting, wasting, and underweight may not match with the original survey as our study was limited only to the subsample who provided information on taking supplementation during pregnancy and lactation, as these were the explanatory variables of interest. There was no precise information on what type of dietary supplement was consumed. Precise information on the micronutrients would help better clarify the beneficial role of the supplement. Data were cross-sectional; hence no causal relationship can be established.

## CONCLUSION

Findings of the study conclude that a great proportion of the women in Timor Leste remain deprived of the benefits of current supplementation programs. Significant socioeconomic variations exist in the uptake of dietary supplements that need to be taken into consideration to improve the coverage. A positive association was observed between maternal consumption and supplementary food and nutritional status of children, which reaffirms the previous findings regarding the role of maternal dietary practices on the nutritional well-being of the children. Given the remarkably high prevalence of child malnutrition in the country, more strategic food and nutrition policy planning within the areas of maternal and child health should be targeted. Longitudinal studies are warranted to validate the causal association.

## ACKNOWLEDGEMENTS

Authors would like to express sincere thanks to the DHS program for providing the datasets used in this study.

### Funding

The authors received no funding for this work.

### Competing Interests

The authors declare that they have no competing interests.

### Author Contributions

- Sanni Yaya analyzed the data, contributed reagents/materials/analysis tools, authored or reviewed drafts of the paper, approved the final draft.
- Ruoxi Wang analyzed the data, authored or reviewed drafts of the paper, approved the final draft.
- Shangfeng Tang analyzed the data, prepared figures and/or tables, authored or reviewed drafts of the paper, approved the final draft.
- Bishwajit Ghose conceived and designed the experiments, performed the experiments, analyzed the data, contributed reagents/materials/analysis tools, prepared figures and/or tables, approved the final draft, revision.

## Human Ethics

The following information was supplied relating to ethical approvals (i.e., approving body and any reference numbers):

DHS surveys are approved by ICF International. The data were secondary and were collected from public domain, therefore no additional approval was necessary.

## Data Availability

http://microdata.worldbank.org/index.php/catalog/2992/.

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
