# Peer review of "Intake of supplementary food during pregnancy and lactation and its association with child nutrition in Timor Leste"

_PeerJ, doi:10.7717/peerj.5935_

## Round 0.1 · original submission · Major Revisions

Dear authors,

Your manuscript has been reviewed by several experts in the analyzed topic and they have found scientific merit in your work. However, there are some major revisions which you must apply in a new version of the text.

With respect and warm regards,
Dr Palazón-Bru (academic editor for PeerJ)

·

Basic reporting

Keeping in mind what is written in the following sections, there are aspects of the work that can be improved at the time of publication, as follows:
- In the summary, lines 52, 53 and 54, in the odds ratio values ​​the corresponding confidence intervals should be placed, as has been done in lines 48 and 49.
- In material and methods, line 142, "and hig maternal mortality rates (citations)" should be added, in order to better understand the paragraph.
On line 156, a bibliographic citation must be entered involuntarily in the form (17).
- In line 168 the recommendations of the WHO are named, but the bibliographic citation should be included.
- In results, lines 243 and 244, the 95% confidence intervals should be included.
- The same on lines 260 and 262.
- In table 2 I think that confidence intervals should be placed for each OR and the values ​​of p.
- The same in table 3.
- Finally figures 1, 2 and 3, should be done in a way that were more illustrative, because due to the large number of columns are difficult to understand with the naked eye. For example, and given the great similarity between the data of stunting, underweight and wasting, perhaps join the figures by type of supplementation and eliminate the complementary columns. Or put another way, place only those that took supplements and eliminate those that did not, in its different forms.

Experimental design

The experimental design is very well done, has a very strong statistical base and the number of participants makes the conclusions of the study and its applicability can be extended to the whole country.

Validity of the findings

I already highlighted in the previous paragraph that the results and conclusions drawn from the work I believe are very valid and can be extrapolated to the entire population, so its applicability can be very good when designing campaigns to prevent malnutrition states through the nutritional supplement in mothers and children.

Additional comments

The work is of a high quality in that sense and include:
The large number of the sample (female diada, children) analyzed.
The very idea of ​​monitoring the state of nutrition and growth using the three variables analyzed in relation to a whole series of factors both maternal and children themselves.
The statistical study performed for each of the three variables considered as outcomes.
The consequences obtained from the work itself with respect to its future applicability, since with the data obtained, it is possible to influence the population at greatest risk, with policies aimed at achieving higher percentages of success in monitoring correct feeding during maternity.

·

Basic reporting

Line 222 the confidence interval of the mean shown in Table 1 should be added to the average age of mothers and children and withdraw these results from that table since it only reflects qualitative variables and appear below column% ..

Correct in table 1 the description of the BMI variable since it is wrong.

Experimental design

The authors acknowledge that there is no precise information on what type of supplementary diet was consumed, but the authors should indicate in the methodology section what is the usual diet of mothers in Timor Leste and what most mothers consider to be dietary supplements.

Validity of the findings

No comment

Additional comments

No comment

·

Basic reporting

Please see the section "General comments for the author".

Experimental design

Please see the section "General comments for the author".

Validity of the findings

Please see the section "General comments for the author".

Additional comments

Reviewer’s comments
Influence of supplementary foods and nutrients on the pregnancy and newborns have been well researched and documented. In the current manuscript, authors have analyzed data of “Timor Leste Demographic and Health Survey (TLDHS)” conducted in 2009-10 to see the effects of supplementary foods (information of the types of supplements is missing) on the health of children. Authors conclude that odds of healthy children in women who take the supplementary foods during the pregnancy and lactation are higher. Manuscript is well written, and data is thoroughly analyzed. But, the manuscript does not provide any new knowledge in terms of contribution of supplementary foods on the health of the children. All the interpretations have been extensively studied previously. I am afraid to say that the manuscript is not suitable to be accepted as research article in the PeerJ.

·

Basic reporting

In general, the manuscript is well written and complies with the review criteria of PeerJ. The most important issue is how data is presented in the figures (see below), all other comments are minor issues.
Specific comments:
Line 89: There is a coma before the parenthesis.
Line 102: “… access to appropriate food, and diverse…”
Line 125: Please define “DHS”
Line 128: “… as there is no currently no research…”
Line 156: There is a “[17]” at the end.
Line 171: “… protein and energy, or frequent infection.”
Line 179: It says “… and not wasted”, it should say “… and no underweight”
Line 207: I suggest expressing the level of significance with the widely used and understood p-value (p<0.05) instead of a percentage.
Line 211: “Stud” should be “Study”.
Line 284: “absolutely” should be “absolute”
Line 297: “and” should be “of”
Line 308: “… in contrast to past evidences.”
Line 310: “… to reduce the the risk of …”
Line 312: “… during pregnancy on birthweight …”
Line 319: “… potential indication of poor maternal healt …”
Figures: Even though the figures are neatly constructed, they do not present the data in a manner easy to understand and interpret. The percentages shown are confusing because they do not relate to the total of children in each category (prevalence). For example, in Figure 1, it is shown that 77.1% of stunted children in urban houses come from mothers that took supplemental food during pregnancy and 22.9% come from mothers that did not (100% is all the stunted children in urban houses, but it should be all the children in urban houses, stunted and not stunted). Since the bars are taller for the ones that took supplements, it looks like there are a lot more stunted children from women that took supplemental food than from women that did not. I suggest presenting the data as percentages of the total of children in each category (prevalence). Also, variability of the data should be represented (standard deviation or error bars) and significant differences should be clearly indicated within the figure.
Tables 2 and 3: I suggest highlighting the significant differences instead of the not significant ones.
Throughout the text:
- There is a missing space between the last word and the parenthesis of most references (it seems to be a problem of the references software used).
- All abbreviations should be explained the first time used. Please define HFA, WFH, WFA, CIs, ORs, and others.

Experimental design

- I agree with the author in that the results may vary depending on the type of supplement women take during pregnancy and/or lactation, so it would be good to specify what was considered as “supplementary food”. Do you know if all women in the study took the same or similar supplements?
- In the methods section, it would be good to state if the dietary supplements are provided free of charge to all pregnant women in Timor Leste (to rule out accessibility as a variable). If so, maybe it is possible to know the type of supplement provided to the women in the study.
- How do you calculate prevalence? It looks like Figures are not representing prevalence as it is usually calculated (number of cases divided by total population).

Validity of the findings

In the discussion, it would be good to comment on the possible reasons of the findings, for example: it was clear from the study that women in urban areas are more likely to take supplements, could that be related to how the supplement is distributed? Maybe it is more difficult for women in rural areas to have access to the supplements, henceforth they are less likely to take them.

Additional comments

Overall the manuscript shows considerable promise, it is well written and the data is robust. However, the figures need to be modified to present the data in a manner easy to understand and interpret, according to the results highlighted in the text, as well as clearly indicate statistical differences. Also, there are some minor revisions that would improve the quality of the publication if fixed.

I recommend accepting this manuscript for publication IF the authors revise and improve the quality of the figures of the manuscript, considering the comments above.

---

## Round 0.2 · accepted · Accept

Dear authors,

I am pleased to inform you that your paper has been accepted for publication in its current form in PeerJ.

Congratulations!

With respect and warm regards,
Dr Palazón-Bru (academic editor for PeerJ)

·

Basic reporting

The authors have satisfactorily completed all the suggestions made by the reviewers, so I believe that the work is correct and can be published.

Experimental design

The authors have satisfactorily completed all the suggestions made by the reviewers, so I believe that the work is correct and can be published.

Validity of the findings

The authors have satisfactorily completed all the suggestions made by the reviewers, so I believe that the work is correct and can be published.

Additional comments

The authors have satisfactorily completed all the suggestions made by the reviewers, so I believe that the work is correct and can be published.

·

Basic reporting

no comment

Experimental design

no comment

Validity of the findings

no comment

Additional comments

Authors have justified my previously raised issues. Improved manuscript is well suited for the acceptance.

·

Basic reporting

The manuscript is well written and has been further improved from its first version, I thank the authors for their hard work.

The figures look better, but they are still a little hard to interpret. I suggest adding the percentage scale in the Y axis.

Experimental design

No comments

Validity of the findings

No comments

Additional comments

No comments